# Acoustic Backscatter Communication and Power Transfer for Batteryless Wireless Sensors

**DOI:** 10.3390/s23073617

**Published:** 2023-03-30

**Authors:** Peter Oppermann, Bernd-Christian Renner

**Affiliations:** Hamburg University of Technology, TUHH, Institute for Autonomous Cyber-Physical Systems, 21073 Hamburg, Germany; christian.renner@tuhh.de

**Keywords:** wireless power transfer, ultrasonics, backscatter communication

## Abstract

Sensors for industrial and structural health monitoring are often in shielded and hard-to-reach places. Acoustic wireless power transfer (WPT) and piezoelectric backscatter enable batteryless sensors in such scenarios. Although the low efficiency of WPT demands power-conserving sensor nodes, backscatter communication, which consumes near-zero power, has not yet been combined with WPT. This study reviews the available approaches to acoustic WPT and active and passive acoustic through-metal communication. We design a batteryless and backscattering tag prototype from commercially available components. Analysis of the prototypes reveals that low-power hardware poses additional challenges for communication, i.e., unstable and inaccurate oscillators. Therefore, we implement a software-defined receiver using digital phase-locked loops (DPLLs) to mitigate the effects of oscillator instability. We show that DPLLs enable reliable backscatter communication with inaccurate clocks using simulation and real-world measurements. Our prototype achieves communication at 2 kBs${}^{-1}$ over a distance of 3 m. Furthermore, during transmission, the prototype consumes less than 300 μW power. At the same time, over 4 mW of power is received through wireless transmission over a distance of 3 m with an efficiency of 2.8%.

## 1. Introduction

Recently, the Internet of Things (IoT) has made tremendous progress in creating a smarter and more efficient infrastructure, driven by the emergence of sensor nodes that are small, cheap, and easy to deploy [1]. These sensor networks rely on low-power communication techniques, such as LoRa, enabling runtimes of several years on a battery or even batteryless sensors driven by energy harvesting [2,3]. However, conventional wireless communication is inapplicable in many scenarios due to electromagnetic shielding. Applications include structural health monitoring, where sensor nodes are embedded within a structure [4,5,6], underwater communication [7,8], fallback systems in nuclear power plants [9], and generally sensors within enclosed metal containers, e.g., pipelines, ship hulls, or cargo containers [10]. In contrast to electromagnetic waves, acoustic waves experience little attenuation in these scenarios and allow efficient transmission over longer distances. Powering these sensors is an additional challenge, as it is often desirable to avoid cable-bound solutions because of high deployment costs in large structures or compromised structural integrity, e.g., in pressurized containers or underwater applications. At the same time, regular battery replacements must be avoided as well because sensors may be in hard-to-reach places. Therefore, several research groups have investigated the transfer of energy and data using acoustic waves through metal, concrete, and water. We sketch a through-metal sensor network for a potential application, pipeline monitoring, in Figure 1.

In through-metal acoustic communication and power transfer, we distinguish between two types of channels: the Sandwich-Plated Channel (SWP), where the two transducers are placed coaxially on opposing sides of a metal boundary, and the Guided-Wave Channel (GWC), where the transducers are placed at different locations on a plate-like metal structure [11]. In the former case, distances are usually only a few centimeters, and bulk waves with relatively high bandwidths may be used. The SWP has been modeled analytically and numerically, and models are well suited to predict power transfer efficiency [12]. However, in the case of GWC, the transducers may be several meters apart. Guided waves, also called lamb waves, travel with little attenuation over long distances through these structures. Lamb waves are best excited at lower frequencies, significantly reducing the channel’s bandwidth. Several research groups have investigated power transfer, communication, or combined power transfer and communication using lamb waves. However, all of them employed active communication schemes. In many applications, passive backscatter communication is advantageous because active communication requires oscillators, amplifiers, and analog circuitry for signal generation, consuming significant amounts of energy and increasing the size and cost of the sensor node. Consequently, communication modules often require orders of magnitude more power than the sensors employed on the nodes [13]. Therefore, backscatter communication has the potential to reduce overall power consumption significantly.

In other domains, acoustic backscatter communication has been investigated in combination with wireless power transfer, i.e., for underwater [7,8] and through-concrete applications [4]. In our previous work, we have shown that such approaches can also be transferred to lamb-wave-based through-metal communication [11,14]. However, to our knowledge, no fully energy-autonomous implementation has been demonstrated that combines acoustic power transfer and passive backscatter communication through metals. Furthermore, most of the existing studies in the field use expensive and bulky high-power lab equipment, i.e., signal generators and oscilloscopes, to establish a proof of concept. However, additional challenges appear when transferring the concepts to true wireless sensor nodes with a strictly limited energy budget, such as Sampling Frequency Error (SFE) between the reader and tag. Especially in ultra-low-power devices, we must expect inaccurate oscillators and implement countermeasures. Lastly, the existing investigation of wireless power transfer and active communication only covers short distances between the reader and tag—often less than a meter—and only investigates simple, isolated plates or beams. In real applications, we expect larger and more complex structures.

This work bridges the gap between the lab and the real world. We demonstrate how backscatter communication and power transfer can be combined to create a fully batteryless miniaturized tag. Therefore, we synthesize approaches from acoustic power transfer and backscatter communication from other domains and focus primarily on practicability. We design low-power hardware from commercially available off-the-shelf (COTS) components instead of using high-cost and high-power signal generators and oscilloscopes. We implement a harvester and modulator module working on the same transducer to enable cheap and small tag devices. Furthermore, we employ Digital Phase Locked Loop (DPLL)s to cope with inaccurate and unstable low-power oscillators. More specifically, we detect the time-varying phase offsets and track the sample timing using a derivative timing detector. In our evaluation, we achieve combined power transfer and communication over up to three meters and uplink data rates of 2 kbit s${}^{-1}$.

The remainder of this article is organized as follows: Section 2 gives an overview of existing approaches to acoustic wireless power transfer and communication through metals, after which we explain our contribution to the field in more detail. Backscatter communication and the design of our prototype are explained in Section 3. The evaluation in Section 4.1 first evaluates the potential of acoustic power transfer through lamb waves and then moves on to the efficiency of the presented harvester. Section 4.2 compares the ultra-low-power oscillator with a more accurate external one. We then investigate the backscatter communication’s power consumption in Section 4.3 and look at the decoding performance of the software-defined receiver in Section 4.4. Finally, we discuss the results and conclude our work.

## 2. State of the Art

### 2.1. Ultrasonic Energy Transfer

Multiple studies have shown the transmission of power through acoustic waves in GWCs. We summarize the results in Table 1. Unfortunately, these studies cannot be directly compared against each other because they used vastly different setups, i.e., transducers, frequencies, impedance matching circuits, and structures, yielding output powers ranging from less than 100 μW to 470 mW. The following paragraphs explain the variables in which existing studies vary.

Piezoelectric transducers come in many different shapes. The most commonly used are disk-shaped and generate a circular wave, propagating in all directions uniformly. In contrast, rectangular transducers may generate directed waves, increasing efficiency at the cost of generality [15]. Some research groups have used transducer arrays to enable beam formation and thus dynamically concentrate the emitted energy in the direction of a specific receiver [16].

Impedance matching at the transmitter and receiver is critical to achieving high efficiency, e.g., Xu [17] reports a 6.75-times increase in output power when using an ideal complex impedance matching circuit at the receiver’s transducer. Unfortunately, in GWCs, the input and output impedances of the transducers are coupled to the structure’s mechanical impedance and vary strongly—even switching between inductive and capacitive behavior—when the carrier frequency changes by only a few hundred hertz. As a result, the optimal matching impedance is not known prior to transducer deployment. Therefore, optimal complex impedance matching requires the design of a dedicated impedance matching circuit after deployment, complicating the setup of such systems, especially in hard-to-access places. Alternatively, the receiver’s Power Management Unit (PMU) can perform Maximum Power Point Tracking (MPPT), where the circuit automatically matches the resistance of the transducer but not the reactance. Furthermore, in some studies, the carrier frequency is automatically tuned to maximize received power, thereby avoiding strong impedance mismatches and improving transmitted power by more than 15 dB over a fixed carrier frequency [5].

When reporting transmission efficiency, it is essential to mention the scope of the measurement. While some studies report the DC output power the harvester provides after rectification and voltage regulation, other studies measure the AC power available at the receiving transducer. Rectification and voltage regulation can be very inefficient and strongly influence the overall Wireless Power Transfer (WPT) system’s efficiency. Another difference considers the presence of a storage element. Although the harvester’s output can be used directly to power the sensor node, transferring the harvested energy into a battery or supercapacitor for later use is also possible. Using a storage element may enable harvesting from sources that provide very little power. The harvester can collect energy and only activate the sensor node once sufficient energy has been harvested. Interestingly, all the reported studies only tested short channels of less than 1 m in length. However, ultrasonic lamb waves can propagate long distances with little loss, as their power is attenuated only linearly with increasing distance. Therefore, we expect much longer distances than what are currently reported.

**Table 1 sensors-23-03617-t001:** Summary of existing studies on lamb-wave-based WPT through metal channels.

Study	Distance m	Frequency kHz	Output Power mW	Efficiency	Impedance Match	Scope ^1^	Storage
Xu, 2022 [17]	0.40	150	1.54	0.217%	Ideal	DC	Yes
Tang, 2022 [5]	<0.30	400	0.08	-	MPPT	DC	Yes
Sun, 2021 [18]	0.40	150	3.18	-	Ideal	AC	No
Tseng, 2020 [15]	0.24	25	470.00	56%	Ideal	AC	No
Shaik, 2020 [16]	0.50	250	0.18	-	MPPT	DC	Yes
Kiziroglou, 2017 [19]	1.00	48	18.00	0.1%	Resistive	AC	No
Kural, 2014 [20]	0.54	220	17.00	30%	N/R	AC	No
**This work**	3.00	225	4.00	2.8%	MPPT	DC	Yes

^1^ DC: Efficiency was measured, including rectification and voltage regulation.

### 2.2. Through-Metal Communication

Through-metal acoustic communication was thoroughly investigated in several different scenarios. We distinguish them by two criteria: channel type and signal generation. Many studies investigate SWP, where transducers are aligned coaxially on opposite sides of a metal barrier. Bulk waves are used in this channel type, and distances are usually small (in the range of a few centimeters). In contrast, lamb waves are used in GWCs that travel along a thin metal structure, e.g., a steel beam, a pipeline, or a ship hull. Such waves can travel over long distances with little attenuation. However, lamb waves are generated efficiently only in low-frequency ranges, limiting communication bandwidth severely. At the same time, existing studies can be distinguished by active and passive communication. Although passive communication has been studied extensively within SWP [21], all previous work on GWC used active communication schemes, which require active circuitry for waveform generation and amplification at the tag node. Instead, passive communication requires essentially only a switch to modulate the transducer’s load impedance, thereby changing its reflection coefficient. This enables smaller, cheaper, and more energy-efficient sensor nodes. In this work, we focus on passive GWC communication but give an overview of existing active approaches for comparison in Table 2.

A challenge in GWC is strong multipath propagation leading to inter-symbol interference (ISI). Different methods have been developed to mitigate the effects of ISI. Some use the correlation with the channel impulse response (IR), where the receiver measures the IR first and then correlates every received pulse with the recorded IR. Other methods use the so-called time-reversal technique [24], where the impulse response is measured and known to the transmitter, which uses the inverse of the IR as the transmit pulse. Both approaches have disadvantages, as the first is computationally very complex on the receiver side, and the second requires the sender to be able to generate arbitrary waveforms, therefore requiring more computational power and complex analog circuitry. Lastly, in some studies, the reflections from the plate’s edges are damped mechanically, e.g., by using clay as damper material. Modifying the structure to reduce ISI is often not feasible in real-world scenarios. As with wireless energy transfer, many studies use laboratory equipment as a proof of concept, leveraging high computational power computers for decoding and perfectly synchronized oscillators on the sender and receiver. An exception is Tang et al. [5], who present a CMOS-integrated circuit.

We have demonstrated in previous work that acoustic backscatter communication is also achievable in GWC over distances up to three meters. First, we demonstrated that the backscatter devices can leverage higher-order modulation with up to 16-QAM-like modulation [11], reaching data rates of roughly 2 kbit s${}^{-1}$. The major limitation for higher data rates, however, is the strong ISI in GWCs. Our follow-up work presented and analyzed a method to equalize the nonlinear GWC, successfully mitigating ISI and increasing achievable data rates to more than 10 kbit/s [14].

### 2.3. Contributions

While some of the work in Section 2.1 also includes communication from the sensor node back to the generator [5,16,18], none of them leveraged backscatter communication to enable even more energy-efficient sensor nodes. Our previous work showed the possibility of fast acoustic backscatter communication but did not employ wireless energy transfer to achieve completely batteryless sensor nodes. Furthermore, our prototype was not implemented energy-efficiently, so we did not investigate energy requirements for the backscatter communication. Moreover, the study was limited to a small, isolated specimen with a maximum distance of 3 m between the reader and tag.

In this work, to cover larger communication distances in more realistic structures, we employ a method to reduce the reader’s self-interference, which arises because the reader does not solely receive the backscattered waveform from the tag but also many other reflections, e.g., from the edges of the structure. When the tag is far away from the reader, the backscattered signal is much weaker than the self-interference, and the reflected signal can not be isolated. To extend communication ranges, the backscattered signal is shifted in frequency to distinguish it from the self-interference. We analyze the potential of WPT over longer distances and design an efficient harvester. Detailed investigation of harvestable power and power consumption of the tag node is provided, showing that WPT can easily exceed the distances reported in prior studies.

Furthermore, we tackle the challenges linked with real-world implementations on low-cost and low-power Microcontroller Unit (MCU)s. Low-power oscillators on such devices are inaccurate and vary in frequency over time. Moreover, reusing the same transducer for harvesting as for communication introduces another source of interference to the backscatter communication. However, employing equalization for ISI mitigation requires accurate channel estimation and is highly sensitive to such time-varying interferences.

## 3. Materials and Methods

This section first describes the design of our hardware prototype and explains critical design decisions. We give an architecture overview in Figure 2. We first focus on the energy harvesting unit at the tag and then move on to the backscatter modulator. After that, we describe the reader circuit and software-defined receiver pipeline that mitigates the effects of ISI by equalization and corrects carrier frequency and timing offsets, thereby enabling communication with inaccurate low-power oscillators in tag devices.

### 3.1. Energy Harvesting

The energy harvesting circuit collects energy from the tag’s transducer and provides a regulated supply voltage sufficient to run the MCU. Figure 3 provides a schematic overview of the circuit. In contrast to most existing studies, we use the same transducer for harvesting and backscattering, which saves space and costs. As we test the limitations of WPT in metal channels, we want to design a harvester that functions in a broad range of scenarios. That poses three challenges, namely
The voltage on the receiving transducer is likely smaller than the required supply voltage for the MCU and potential sensors,The maximum power that can be extracted from the piezo may not be sufficient to power the MCU continuously,The transducer’s output impedance may vary strongly. Therefore, the harvester needs to work efficiently with a broad range of impedances.

The first challenge can be overcome using a voltage booster circuit, the second requires harvesting energy in a storage capacitor and duty-cycle of the MCU, and the third requires a load adaption mechanism, e.g., MPPT.

A flexible solution to all three problems is to use a commercially available PMU IC. We decided to use the ADP5091 from Analog Devices because it has an inbuilt boost regulator that can boost input voltages as low as 80 mV. However, in cold start, a minimum voltage of 300 mV is required. To fulfill this cold start threshold voltage, we first amplify and rectify the AC voltage from the transducer with a voltage quadrupler circuit. The PMU also supplies duty cycling through an output pin that signals if the voltage on the charge element has reached a configurable threshold. Furthermore, it can efficiently manage stored energy and prevent damage to the storage element by only charging it up to a configurable voltage and not discharging it below a certain voltage.

Additionally, the PMU performs MPPT with a configurable voltage threshold. MPPT is a technique to adapt the resistive load of the harvester dynamically. The PMU realizes MPPT by regularly stopping harvesting, sampling the open circuit voltage at the input capacitor, and then only extracting energy from the input capacitor $C_{\mathrm{IN}}$ to the storage capacitor $C_{\mathrm{ST}}$ when the voltage exceeds the configurable MPPT threshold. We investigate the optimal MPPT threshold in Section 4.1.1. In general, the piezoelectric source has a complex-valued internal impedance, and for maximum power transfer, and the harvester’s load impedance has to be matched with a complex matching circuit between the transducer and harvester. However, the complex impedance strongly depends on the structure, carrier frequency, and exact position of the transducer on the structure. Therefore, designing exact impedance matching circuits is a task that has to be manually repeated for every sensor node [17], which is highly impractical. Instead, we approximate the impedance only by adapting the resistive load.

Once the storage capacitor exceeds a terminal voltage $V_{\mathrm{th}+}$, the PMU provides a turn-on signal, which is called PGOOD. A hysteresis is programmed so that PGOOD stays high until the system voltage drops below a second threshold $V_{\mathrm{th}-}$. We use this signal to enable a low-power LDO regulator that provides the regulated supply voltage for the MCU. Setting the size of $C_{\mathrm{ST}}$, and the threshold voltages, the harvester can be configured to supply a wide range of applications with different power requirements independent of the input power provided by the transducer through duty cycling. In the presented prototype, we set $C_{\mathrm{ST}}=200$ μF, $V_{\mathrm{th}+}=5$ V, and $V_{\mathrm{th}-}=2$ V. Neglecting losses, the buffered energy supports an output power of 2.1 mW for one second, enough to take a sensor reading and transmit the result. A detailed evaluation of the harvesting performance is given in Section 4.1.

### 3.2. Backscatter Modulation

Backscatter communication leverages the fact that an impedance mismatch between the transducer’s output impedance and the circuit’s input impedance causes the reflection of some of the incoming power. The complex reflection coefficient Γ is defined as
(1)$$\Gamma \left(z_{L}\right)=\frac{z_{L}-z_{T}}{z_{L}+z_{T}},$$
where $z_{L}$ is the load impedance applied to the transducer and $z_{T}$ is the transducer’s internal impedance. It describes the magnitude fraction and a phase shift of the reflected signal relative to the incoming signal. The tag circuit shown in Figure 4 utilizes this fact to change its reflection coefficient systematically, thereby modulating information on the reflected waveform. Accordingly, to change its reflection, the backscatter modulator has to vary the load impedance connected across the transducer’s terminals. The transistors allow short-circuiting of the load at the transducer by pulling the transistor gate up, while the load impedance is very large when the gate is pulled down. These two states will effectively lead to a reflection coefficient of $\Gamma_{0}=\Gamma \left(0\right)\approx -1$, and $\Gamma_{1}=\Gamma \left(\infty \right)\approx 1$, yielding a phase shift of the reflected signal of 180°.

However, the harvester module is connected in parallel with the modulator. When the transistors in the modulator circuit block current flow between the transducer terminals, the whole circuit’s impedance is dominated by the harvester’s input impedance. Accordingly, the harvester influences the reflection coefficient in this state and might vary depending on the charge of the storage capacitor. Therefore, we employed a switch between the modulation and harvesting circuits, allowing the MCU to completely disconnect the harvester module from the transducer during the communication phase. However, this requires a sufficient energy buffer to power the MCU while the harvester is disconnected.

In previous work, we switched between different load impedances on the tag, where a specific impedance represented a symbol value. In this case, the transmission band is centered around the carrier frequency $f_{c}$, and the reader can not isolate the backscattered signal from its self-interference, i.e., other reflections from the environment. In that case, the reader can not detect the backscattered signal when it is weak compared to the self-interference. In order to amplify the reflected signal before sampling, the reader must separate it from the self-interference. As also presented in [8], we shift the transmission band away from the carrier frequency $f_{c}$ by mixing it with an intermediate frequency $f_{i}$

The MCU generates the digital square wave (sq(t)) with frequency $f_{i}$ using its pulse width modulation (PWM) peripheral. The square wave signal is applied to the transistor gates, effectively switching the reflection coefficient between 1 and -1. We can write the reflection coefficient as
(2)$$\Gamma \left(t\right)=\mathrm{sq}\left(\omega_{i}t+\phi_{t}\right)=\frac{4}{\pi }\sum_{k=1}^{\infty }\frac{\sin\left(\left(2k-1\right)\omega_{i}t+\phi_{t}\right)}{2k-1},$$
where $\omega_{i}=2\pi f_{i}$, and $\phi_{t}$ is the phase of the square wave. The last equation stems from the well-known spectrum of a square wave, which has an infinite amount of frequency components at odd multiples of its base frequency. The symbols are then modulated by shifting the phase of the square wave ($\phi_{t}$), effectively implementing phase-shift keying (PSK) modulation.

The backscattered signal s(t) is the product of the incoming carrier and the reflection coefficient, i.e.,
(3)$$s\left(t\right)=a_{0}\sin\left(\omega_{c}t+\phi_{0}\right)\;\cdot \;\Gamma \left(t\right),$$
with $\omega_{c}=2\pi f_{c}$. Reader and tag are not synchronized, leading to an arbitrary face offset $\phi_{0}$ between the carrier and the intermediate frequency. Furthermore, the amplitude of the incoming carrier $a_{0}$ depends on the channel between the reader and tag. Using the trigonometric identity
(4)$$\sin\left(a\right)\;\cdot \;\sin\left(b\right)=\frac{1}{2}\left[\cos(a-b)-\cos(a+b)\right],$$
we can rewrite Equation (3) as
(5)$$s\left(t\right)=\frac{4a_{0}}{2\pi }\sum_{k=1}^{\infty }\frac{\cos\left(\left(\omega_{c}-\left(2k-1\right)\omega_{i}\right)t+\phi_{0}-\phi_{t}\right)-\cos\left(\left(\omega_{c}+\left(2k-1\right)\omega_{i}\right)t+\phi_{0}+\phi_{t}\right)}{2k-1},$$
showing that the backscattered signal does not contain a frequency component at $f_{c}$.

The shift of the transmission frequency enables the reader to remove the self-interference at $f_{c}$ before amplification and sampling, leaving only the reflected signal in an infinite amount of sidebands around $f_{c}+\left(2k-1\right)\;\cdot \;f_{i}$ and $f_{c}-\left(2k-1\right)\;\cdot \;f_{i}$, where $k\in N^{+}$. The whole process is sketched in Figure 5.

Since the backscatter modulation—the switching of the transistors—does not require much power, the MCU is the dominant contributor to total power consumption. It requires running hardware timers and sufficiently fast clocks to generate the square wave. Thus, power-aware implementation of the tag node can greatly reduce the required energy buffer and charging times. For CMOS technology, the dynamic power $P_{\mathrm{dyn}}$ that an MCU consumes scales quadratically with the supply voltage *V* and linearly with the MCU’s clock frequency $f_{\mathrm{mcu}}$, i.e.,
(6)$$P_{\mathrm{dyn}}\propto f_{\mathrm{mcu}}\;\cdot \;V^{2}.$$
Therefore, we select a dedicated low-power MCU (STM32L073RZ) and leverage all its power-saving features. This includes a reduced supply voltage of only 1.8 V, saving nearly 70% of power consumption compared to 3.3 V. Furthermore, choosing the internal low-power oscillator and scaling the clock frequency to the minimum requirement is advantageous. As a downside, the on-chip low-power oscillators are unstable and inaccurate, introducing an SFE that must be detected and compensated at the receiver. We will explain this in Section 3.4.

### 3.3. Carrier Generation and Signal Reception

The reader hardware fulfills two functionalities. It generates a carrier with a customizable frequency and receives and demodulates the reflected signal from the tag. A schematic overview is given in Figure 6. We use a dedicated waveform generator IC for signal generation, the AD9833 from Analog, which allows fine-tuning of the carrier frequency with up to 20 mHz resolution. The generated sine wave is amplified to a 30 V peak-to-peak amplitude to drive the piezo transducer.

The same transducer is used for continuous signal generation and reception. We measure the current through the transducer to receive the reflected signal. A small 10 Ω shunt resistor is placed in series with the transducer so that the voltage drop over the transducer is proportional to the current. After filtering, an analog demodulator removes the high-frequency carrier at $f_{c}$, leaving only the backscattered signal at the intermediate frequency $f_{i}$. The demodulation of the intermediate frequency is performed in software and is described in Section 3.4.

Commercially available demodulator ICs are designed to work with RF signals and are unsuitable for frequencies as low as several hundred kHz. Therefore, we construct a demodulator from operational amplifiers and a switch. With the local oscillator as a control input, the switch changes between the received signal and the inversion of the received signal, effectively mixing the received signal with a square wave. After removing the high-frequency contributions of the result with a low-pass filter, the self-interference at $f_{c}$ is shifted to a DC component, and the reflected signal is isolated with a bandpass filter. We provide a detailed mathematical description of the demodulation process in Appendix B.

After filtering and amplification, the MCU samples the resulting signal with the internal analog-to-digital converter (ADC). The amplifier gain can be dynamically set by MCU, thus allowing the reception of a wide range of signal strengths.

After introducing the modulation and demodulation scheme and the prototype hardware, we want to clarify the source of the SFE in Figure 7. In a conventional radio, Carrier Frequency Offset (CFO) and SFE are two separate effects, as the carrier is usually generated from a dedicated RF clock, while the baseband signal is generated from a digital device with a dedicated clock source. In our backscatter prototype, the reader generates the carrier frequency and uses the same clock for demodulation. Therefore, no CFO arises at this stage. However, the tag uses its oscillator for the generation of the intermediate frequency $f_{i}$ and for symbol timing, and this clock deviates from the sampling clock at the reader. This offset produces a symbol timing error and a phase rotation in the sampled signal. Although the phase rotation is introduced by a mismatch of the intermediate frequency, its effect is the same as from carrier frequency mismatch. Therefore, we will still refer to it as CFO for the remainder of this work.

### 3.4. Packet Structure

Our packet structure consists of three parts: First, a synchronization sequence, where the tag generates the same symbol for multiple symbol periods. This sequence is designed to be long enough to yield a steady state despite the Inter Symbol Interference (ISI), allowing the receiver to roughly estimate the CFO and signal amplitude required for robust preamble detection. The second part is the preamble, where we employ a 13-symbol barker sequence. Finally, the actual symbols are sent. Different modulation orders may be used to achieve higher data rates or more robust signal detection, depending on the SNR of the channel. In this work, we have only investigated 4-PSK as a modulation scheme.

#### Software-Defined Receiver

After sampling, the signal is transformed into a baseband in software by digitally multiplying it with a sine of the same frequency. By choosing a sampling rate of $4f_{i}$, the transformation to complex baseband samples with an in-phase component $y_{I}$ and a quadrature component $y_{Q}$ is
(7)$$\begin{array}{cc} y_{I}\left[i\right] & ={\left(-1\right)}^{i+1}y\left[2i\right] \end{array}$$
(8)$$\begin{array}{cc} y_{Q}\left[i\right] & ={\left(-1\right)}^{i}y\left[2i+1\right]. \end{array}$$
Finally, this resulting signal has a sampling rate of $2\;\cdot \;f_{i}$ and can be further downsampled to a multiple *L* of the symbol rate $f_{m}$.

Due to the previously described clock instability, three errors may arise in the baseband signal—CFO, Phase Offset (PO), and STE. We apply standard techniques from software-defined radio receivers, using DPLLs to track and correct these errors. The whole reception process consists of the three stages shown in Figure 8. First, in the *carrier detection stage*, the receiver detects the presence of the signal by a signal power threshold. Once the threshold has been exceeded, this stage also uses the recorded samples to perform a rough CFO and signal amplitude estimation. The CFO estimate ${\hat{f}}_{\mathrm{CFO}}$ is important because the preamble can not be detected reliably when the CFO is large. At the same time, the signal is scaled to unity power for each message, as the optimal gains and parameters for DPLLs in subsequent stages depend on the signal amplitude $\hat{a}$. For the detection and compensation of timing errors, we need to oversample the signal with a multiple *L* of the symbol rate.

In the *preamble detection* stage, the receiver corrects the incoming samples by phase-shifting and scaling them with the estimated errors. It then searches for the start of a message by correlating it with the preamble. Once a peak in the correlation exceeds a fixed threshold, the start of the message is detected. In the following *symbol decoding* stage, the DPLLs track the phase and timing errors during the message. The detected preamble is used to estimate the initial CFO and STE α for DPLL initialization. Second, we also use the preamble to train the equalizer. As the preamble is used for parameter estimation and equalizer training, its length is crucial. A longer preamble increases the accuracy of the estimated parameters and equalizer weights but also adds overhead and keeps the tag active without transmitting any payload.

In the last stage, the data symbols are decoded. At the input of the stage, the signal is scaled and phase-shifted to compensate for CFO and PO. Then an interpolator corrects the STE. The samples are subsequently equalized with a decision feedback equalizer to remove ISI. Finally, a symbol decision is performed. After each symbol, the timing and phase error DPLLs are updated.

The DPLLs each consist of an error detector and a loop filter. The exact phase error of the *k*-th symbol $\delta_{k}$ is
(9)$$\delta_{k}=∠\left(a_{k}\right)-∠\left({\hat{a}}_{k}\right),$$
where *∠* denotes the phase angle of a complex number. To avoid computing the angle explicitly—which is computationally expensive—we approximate the phase error by
(10)$$\delta_{k}\approx ℑ\left\{a_{k}\;\cdot \;{\hat{a}}_{k}^{*}\right\},$$
where ${\hat{a}}_{k}^{*}$ is the complex conjugate of the symbol decision. With the PSK constellation points on the unit circle and assuming small errors, the approximation is valid as
(11)$$ℑ\left\{a_{k}\;\cdot \;{\hat{a}}_{k}^{*}\right\}=\left|a_{k}\right|\sin\left(∠\left(a_{k}\right)-∠\left({\hat{a}}_{k}\right)\right)\approx ∠\left(a_{k}\right)-∠\left({\hat{a}}_{k}\right).$$
The loop filter for the phase DPLL is an integrator with a filter gain $g_{p}$. After every symbol, the estimate of the total phase error ${\hat{\phi }}_{k}$ is updated as
(12)$${\hat{\phi }}_{k}={\hat{\phi }}_{k-1}+g_{p}\;\cdot \;\delta_{k}$$

Analogously, the error for the sample timing error DPLL is computed using a derivative timing error detector. The timing error for the *k*-th symbol, $e_{k}^{t}$, is approximated as
(13)$$e_{k}^{t}=ℜ\left\{{\dot{y}}_{Nk}\;\cdot \;y_{Nk}^{*}\right\}\approx ℜ\left\{\left(y_{Nk+1}-y_{Nk-1}\right)\;\cdot \;y_{Nk}^{*}\right\},$$
where ${\dot{y}}_{Nk}$ is the slope of the signal of the *k*-th sample, and *N* is the number of samples per symbol. We provide a detailed explanation of the derivative timing error detector in Appendix C. The loop filter is again an integrator with filter gain $g_{t}$, such that the update of the interpolator input $\beta_{k}$ is computed as
(14)$$\beta_{k}=\beta_{k-1}+g_{t}\;\cdot \;e_{k}^{t}.$$

## 4. Evaluation

### 4.1. Wireless Energy Transfer

This section first analyzes the potential for acoustic energy transfer through GWC over several meters. We establish the maximum power point for the piezoelectric transducers and then evaluate the extractable power and efficiency over different distances and frequencies. Afterward, the efficiency of the presented harvester device is evaluated.

#### 4.1.1. Harvesting Potential

The input impedance of the harvester must match the transducer’s output impedance to maximize extracted power. The PMU can only match the harvester’s input resistance through MPPT but can not perform complex impedance matching because dynamically adapting complex impedances is hard to realize. In MPPT, the harvester only extracts energy from the source when its voltage exceeds a predefined fraction of its open-circuit voltage. Typically, piezoelectric sources are modeled as voltage sources with a nonzero internal series impedance. In that case, the load should match the source’s internal resistance. We evaluate this assumption by measuring voltages and currents over the piezoelectric source with different load resistances. The results are shown in Figure 9. We see an approximately linear relationship between voltage and current, indicating that maximum power is extracted from the source if the MPPT threshold voltage is half the open-circuit voltage. This result confirms observations from [5,16]. The same experiment was repeated with different carrier frequencies, clearly showing that maximum extractable power strongly varies with frequency. This shows the requirement for a dynamic MPPT mechanism compared to a fixed threshold voltage.

Next, we ask how much power AC we can expect to extract at the sensor node at a certain distance. The selected carrier frequency strongly influences this power level. Frequencies that resonate in the structure enable more efficient power transfer. In Figure 9 (bottom), we show the maximum harvestable powers with optimal resistive matching for a band around the transducer’s resonance from 200 kHz to 250 kHz. At 3 m, the best frequencies provided an output power of 4 mW. At that frequency, the reader has generated an input power of 141 mW, leading to a transfer efficiency of 2.8%. Repeating the measurement for multiple channels with varying distances shows that no clear trend can be observed. The maximum power over a 1 m channel is about 10 mW with roughly 8.3% transfer efficiency. The 2 m channel shows almost identical power levels to the 3 m channel. However, the frequencies that produced the highest output power varied strongly between channels. The results indicate that distance is not the dominant factor determining transfer efficiency. Strong frequency dependency suggests that positive interference patterns are essential. These are influenced by structure geometry and the exact placement of the transducers, which are generally unknown at the time of the system design.

#### 4.1.2. Harvester Performance

In the previous section, we only investigated the maximum extractable AC power at the transducer. However, the harvester has to convert AC into DC power to supply microcontrollers and sensors. In this section, we inspect the efficiency of the harvester only. When considering the harvester’s performance, multiple factors may degrade its efficiency: First, the MPPT tracker of the PMU may mismatch the input impedance of the harvester, therefore, not extracting the maximum available power from the source. Furthermore, harvester self-consumption reduces the energy available to the application. This includes power consumed to boost the input voltage in the PMU and leakage currents from the storage and input capacitors.

We define the efficiency of the harvester as the amount of power flowing into the storage capacitor divided by the amount of power that is theoretically available at the source. We evaluate the input power by applying a sine generator with variable series resistance $R_{s}$ on the inputs of the harvester (see Figure 10a). With knowledge of the current $I_{H}$ through and voltage drop $V_{H}$ over the harvester, the power uptake is determined. Furthermore, the maximum available power to the harvester $P_{\max}$ is, according to the maximum power transfer theorem,
(15)$$P_{\max}=\frac{V_{\mathrm{in},\mathrm{eff}}^{2}}{4\;\cdot \;R_{s}}=\frac{V_{\mathrm{in}}^{2}}{8\;\cdot \;R_{s}},$$
where the sinusoidal source provides the effective (RMS) voltage $V_{\mathrm{in},\mathrm{eff}}$. Finally, we measure the harvested power $P_{h}$ from the storage capacitor voltage
(16)$$P_{h}=\frac{\Delta E}{\Delta t}=\frac{C_{\mathrm{ST}}\left(V_{\mathrm{ST}}^{2}\left(t+\Delta t\right)-V_{\mathrm{ST}}^{2}\left(t\right)\right)}{2\;\cdot \;\Delta t},$$
where ΔE is the change in energy stored in the storage capacitor over a short time period Δt. During this evaluation, no MCU was connected to the harvester, so all the measured power consumption is self-consumption from the harvester itself.

Figure 10b shows the voltage on the storage capacitor over time for different source resistances and input voltages. The time required to fill the storage capacitor up to the maximum voltage varies with available power from the source, and if the source does not provide sufficient power, the storage voltage never reaches its terminal voltage but plateaus at a level where the harvested power and self-consumption are in equilibrium. Figure 10c provides a more detailed analysis of the harvested power. In the beginning, when the harvester is in cold start, it takes up about 239 μW, 87% of the maximum available power from the source. That indicates that the impedance match between the source and harvester is not optimal, i.e., the MPPT is not matching the source’s resistance perfectly. Furthermore, considerably less power, i.e., 115 μW, ends up in the storage capacitor. The PMU consumes the remaining power itself. Once the source is turned off, the harvester loses up to 200 μW due to leakage and the quiescent current of the PMU. The self-consumption shown in Figure 10d is highly dependent on the voltage on the storage capacitor, indicating that the leakage current is the dominant factor.

In the performed experiments, the efficiency of the harvester is around 40% to 50%. Exact efficiency is highly variable, as it depends on the power available at the source and even on the voltage over the storage capacitor. These observations are consistent with previously reported harvester efficiencies with comparable input powers, e.g., as reported in [1]. When available power is reduced to only tens of microwatts, the efficiency shrinks to only several percent, as the harvester consumes most of the available power itself. Furthermore, the 300 mV nominal minimum turn-on voltage of the PMU is not a significant limitation. Typical piezoelectric sources have large internal series resistance; hence, higher voltages are required to provide sufficient power for the MCU. However, further optimization of the PMU might allow less leakage and enable more effective harvesting from low-power sources. When the harvester must work with sources providing very little power, the charging threshold at the storage capacitor could be reduced, and a larger capacitor should instead be chosen. A disadvantage of this strategy is that a larger cap increases the charge-up time required until the cap voltage exceeds the minimum required voltage for the MCU.

### 4.2. Communication

This study compares the internal low-speed oscillator integrated into the MCU with a clock frequency of 4.196 MHz with a 4 MHz external crystal oscillator. The internal oscillator is very energy efficient, does not require additional space, and enables the MCU to run in its lowest consumption mode. Its drawback is that it is less accurate and more unstable than the external crystal, i.e., the clock frequency includes significant jitter. In this section, we first investigate the jitter and deviation of the internal and external clocks and measure their influence on tag power consumption. We then evaluate how clock instability affects communication performance and if the presented mitigation strategy is effective.

#### Low-Power Clock Stability

To evaluate the clock stability, we record the square wave produced by the tag’s MCU with an oscilloscope and compare it with an ideal square wave with the nominal carrier frequency. Figure 11 shows the measured CFO over time. We see that the MCU’s supply voltage has a significant impact on the CFO of the internal oscillator, which is roughly 70 Hz larger—more than 800% increase—at $V_{\mathrm{DD}}=1.8$ V compared to 3.3 V. Additionally, even with constant supply voltage, the instantaneous CFO varies throughout a single message by more than 5 Hz. In theory, when using a 4-PSK modulation scheme, a phase offset of up to 45° can still be decoded correctly, allowing an instantaneous CFO of 5 Hz for up to 20 ms. However, since we have to use a nonlinear equalizer to mitigate the strong ISI, already much smaller phase offsets lead to large errors in the equalizer output. Therefore, when using the internal oscillator, a receiver can not rely on a single estimate of the CFO, e.g., during the preamble but must dynamically track and correct phase offsets during message reception. Furthermore, as the tag generates the CFO from its main clock, the same deviation also affects symbol timing. Therefore, symbol timing must also be recovered and tracked throughout message reception. For comparison, we also measure the CFO when using the external crystal. The instantaneous CFO is much more stable and only varies less than a tenth of a hertz during a message transmission. Therefore, when using the external clock, PLLs are not necessarily required during message reception when the static CFO is estimated accurately once at the beginning of the message.

### 4.3. Tag Power Consumption

The tag’s power consumption is important and determines the minimum required energy buffer size. Less energy usage for data transmission enables shorter harvesting times. The consumed energy can be distinguished in PMU self-consumption, static consumption from the MCU for driving its clock and the peripherals, and dynamic consumption for modulating the backscatter signal. The harvester’s self-consumption was analyzed in Section 4.1.2. Once the cap voltage exceeds the charging threshold, the MCU is powered, dominating the total power consumption. We measure the supply current of the MCU to determine its power consumption in stop mode (inactive) and during message transmission. We also evaluate the dynamic consumption required to modulate the backscatter signal.

The MCU provides various power-saving features, i.e., several low-power modes and clock frequency scaling. To reduce the consumed current as much as possible, we supply the MCU at only 1.8 V, scale the clock frequency down to 1 MHz, and use the STOP mode, in which the clock and most peripherals are halted, whenever possible. However, timers and a pulse-waveform generator peripheral are running during the message transmission, preventing STOP mode. Figure 12 shows the smoothed power consumption of the tag during the transmission of one message. It can be seen that the tag consumes only about 6 μW when the MCU is in STOP mode, but total consumption rises strongly during message transmission. When directly comparing internal and external oscillators at 4 MHz, we can observe that usage of the external crystal consumes an additional 500 μW, an increase of 80% compared to the internal oscillator. Additionally, the external oscillator requires a startup time of at least 2 ms after wakeup, and power demand spikes at oscillator startup. Given the high power requirement, we conclude that using the internal oscillator is desirable. Furthermore, when configuring the internal oscillator to only 1 MHz, which is still sufficient for backscattering data at 2 kbit s${}^{-1}$, the MCU consumes less than 300 μW.

We also measured the consumption with and without the modulator circuit connected. The difference in power consumption reveals the dynamic power required for backscattering, i.e., switching the transistor’s gate voltage. The actual backscattering only consumes about 20 μW. Hence, the mostly idling MCU consumes the majority of the power. These results show that it is feasible to perform communication with ultra-low power consumption suitable for wirelessly powered systems using general-purpose MCUs and off-the-shelf components.

### 4.4. Backscatter Communication

We evaluate the communication reliability by first comparing the communication driven by the stable and accurate external crystal oscillator with the low-power internal oscillator. Afterward, we look at the error rates of the backscatter communication in different channels. The setup for the following evaluations is a 3.5 m long beam with several attached piezo transducers. The dimensions and channels used are shown in Figure 13. The transducers are disk-shaped with a diameter of 1 cm and a thickness of 2 mm. Their resonance frequency is around 200 kHz. All transducers were attached directly to the metal using epoxy resin.

#### 4.4.1. Sdr Pipeline

We first analyze the characteristics of the backscatter channel in the test structure. Figure 14 shows in the top graphs the SNR for different carrier frequencies. It can be seen that the SNR fluctuates by more than 20 dB within a few hundred hertz. Remarkably, in our measurements, the distance between the reader and tag transducer was small: the 3 m channel has even higher peak SNRs than the 2 m channel. To provide a detailed evaluation of a diverse range of channels, we pick three distinct carrier frequencies for each distance for further evaluation. In Figure 14 (bottom), the impulse responses for the selected channels are shown. We observe that all channels exhibit extended reverberation for about 10 ms. However, channel impulse responses vary strongly with carrier frequency. Hence, SNR alone is likely unsuitable for estimating a carrier frequency’s communication reliability. The recorded impulse responses are available for download in Appendix A.

Next, we transmitted 250 messages through each channel for each selected frequency and decoded them using our SDR pipeline, as described in Section 3.4, using the internal oscillator at the tag once and the more stable external oscillator once. The modulation scheme was 4-PSK at a symbol rate of 1000 symbols per second, yielding a total data rate of 2 kbit s${}^{-1}$. Each packet contained 200 payload bits.

First, we investigate if the more complex receiver with phase and timing error tracking is advantageous. Therefore, we decode all messages without tracking, i.e., only estimating CFO and STE from the preamble and assuming it is constant throughout the whole message. We then use the DPLLs for tracking the CFO and STE dynamically. The packet error rates (PER) displayed in Figure 15a reveal that nearly all packets are decoded correctly, even without error tracking when using the stable external oscillator. However, nearly 25% of the packets contain errors when using the internal oscillator. DPLLs enable error-free reception of all packets in both cases.

We then compute the SNR of the equalizer output as
(17)$${\mathrm{SNR}}_{\mathrm{out}}=10\log_{10}\left(\frac{\sum_{k}x_{k}^{2}}{\sum_{k}{\left({\hat{y}}_{k}-x_{k}\right)}^{2}}\right),$$
where *x* is the original transmission sequence, and $\hat{y}$ is the equalizer’s output. Figure 14 displays the deviation of output errors. The results show that the instability of the internal oscillator can be mitigated successfully when using the PLLs. The equalizer output’s SNR with the internal oscillator is comparable to the one with the external oscillator. Notably, the results for the different channels are very similar, even though the SNR of the received signal varies by more than 10 dB between the channels. However, when comparing internal and external oscillators, note that the underlying channels are unequal. Even when the same nominal carrier frequency has been configured, the CFO of up to 70 Hz causes the internal oscillator to use a slightly shifted frequency band. This could explain why the internal oscillator yields—despite its instability—a better result than the external one in some channels.

Furthermore, we also evaluated if decoupling the harvester from the modulator circuit improves the signal quality (see Figure 15c). Surprisingly, we did not observe a significant difference between the cases where the harvester is connected and influences the tag’s load impedance and when it is decoupled. Therefore, we conclude that the switch we used for decoupling, described in Section 3.3, is not required.

#### 4.4.2. Simulation Study

In the real-world experiments, we did not have full control over all variables, i.e., the channel impulse response, the extent to which the tag clock varies, and the amount of noise in the signal. To characterize the influence of various parameters on the receiver, we also implemented a channel simulation incorporating multipath propagation, time-varying SFE, and noise levels. In Figure 16 left, we show the influence of different channel impulse responses on the output SNR of the receiver. We compare three channel impulse responses: an ideal channel without ISI and two previously recorded channels. Gaussian noise was applied after convolution with the impulse response to make SNRs comparable between different channels. We observe that the impulse response greatly influences the output of the receiver. With an ideal channel, the SNR of the raw signal and the SNR of the equalized signal are up to 10 dB larger than with impulse response IR1, and channel IR2 produces less noise at the equalizer output than IR1. In Figure 16 right, we generated a gaussian distributed time-varying SFE similar to the observed variations from the actual low-power oscillator shown in Figure 11. We varied the standard deviation of the simulated SFE and measured the Bit Error Rate (BER) over 5000 simulated messages (1 million bits). The results indicate that even with an SFE close to zero, the decoder without PLL has a high BER ($5\times 10^{-3}$). We attribute this high BER to incorrect estimation of the SFE during preamble detection, accumulating into larger phase errors throughout the message. In direct comparison, a decoder with PLL significantly reduces the bit error rate by more than two orders of magnitude to roughly $1\times 10^{-5}$. However, the gain of the PLL plays an important role. With an SFE deviation in the range of $2\times 10^{-4}$ (200 ppm), which we observed on the low-power oscillator, the decoders with medium to high gain PLL achieved BERs of about $3\times 10^{-4}$.

### 4.5. Larger Structures

Prior work—including the test presented in the above sections—was only evaluated on small and isolated structures, e.g., small plates or pipes. In this work, we investigate the potential of wireless acoustic energy transfer on more realistic real-world structures. We, therefore, attached several transducers to a steel beam supporting a crane shown in Figure 17, with transducer distances of 1, 2, 3, 4, and 5 m. We used our reader to generate a carrier wave and measured the signal amplitude at the different transducers.

Compared to the smaller specimen we used in the previous evaluations, the amplitudes are smaller in the large beam. When choosing the resonant frequencies, amplitudes in the large beam reach up to 400 mV over 1 and 2 m distances and still about 300 mV at 5 m. However, the observed amplitudes also suggest that the acoustic waves experience little attenuation when traveling through the metal beam, i.e., a fivefold distance increase from 1 m to 5 m only corresponds to a 25% decrease in amplitude.

Unfortunately, the wirelessly transmitted power over this large structure cannot charge our prototype tag fully. However, several improvements can still enable batteryless tags on such large structures. First, increased sender powercan provide sufficient energy. Second, the used transducers and their frequency band do not fit well to the large beam, with a thickness of more than 2 cm. Larger transducers combined with lower carrier frequencies likely improve the efficiency of the wave excitation strongly. Third, the harvester can also be optimized to work with less available power in a duty cycle, as discussed in Section 4.1.2. In essence, larger structures may require more transmit power by the reader, but once a strong enough wave is excited in a structure, it can cover large distances.

## 5. Discussion

The investigation of WPT through an elongated metal bar showed that more than 4 mW AC power could be transferred over a distance of 3 m, supporting a wide range of sensing applications: Tang et al. [5] report SHM measurements using lamb waves at sub-milliwatt power. Many other sensors require even less power, e.g., low-resolution image recording [7] can be achieved at several hundred microwatts, acceleration measurements at tens of microwatts, strain gauge, temperature, and humidity measurements consume even less. Furthermore, a duty cycle can support even higher-power applications requiring extended computations.

In theory, the signal power of guided lamb waves is attenuated only linearly with distance, enabling much longer distances. Practical measurements on the large structure support the ability to transfer power over large distances, i.e., the maximum observed voltage at the receiver after 5 m was still about 75% of the observed voltage over 1 m. The presented harvester prototype required at least a signal power of 200 μW to fully charge the capacitor, which can potentially be reduced to about 40 μW when cap voltage is reduced to 3 V, as discussed in Section 4.1.2. The results relate well to harvesters presented in RF harvesting, e.g., the harvester presented in [1] showed dramatic drops in efficiency when the energy source only provides several tens of μW. The total efficiency of WPT was 8% over 1 m, which exceeds the observations from [19] (0.1% at 1 m). However, the efficiency is likely strongly related to the structure, the transducer’s geometry, and the bounding method. Therefore, a direct comparison of these results is not informative.

Tailoring the energy transfer method to a specific structure could greatly increase the overall transfer efficiency of less than 3% over 3 m distance drastically. For example, Tseng et al. [15] used a finite element model simulation of the metal waveguide to find its resonance frequencies and chose transducers with matching resonance and determined their optimal placement position to achieve the highest transfer efficiency. Other groups used rectangular transducers to generate directed waves and increase transmission efficiency [20]. However, such optimizations are hard to achieve as real structures often have complex geometry and unknown boundary conditions, prohibiting previous determination of optimal locations and frequencies. We, therefore, aimed to investigate a *plug-and-play* solution that works without prior knowledge about the channel. However, dynamic beamforming using a piezo array may be a promising optimization.

The presented backscatter tag in this study required less than 300 μW during backscattering, of which only 20 μW were required for modulation, and the MCU required the rest. Shaik reports a power consumption of 100 to 1000 μW during active uplink communication, dependent on the data rate. While ASIC implementation is potentially much more energy efficient, we showed that a working system is achievable with commercially available off-the-shelf components. For research prototypes and small series production, this is desirable as ASIC production is very costly and time-consuming. The backscatter communication decoding with DPLLs proved robust against the CFO and STE induced by low-power oscillators. Furthermore, the signal strength over 3 m was more than sufficient due to canceling of the self-interference. The WPT will likely be the limiting factor for longer distances in large structures. However, backscatter communication can potentially be used over much longer distances in battery-powered sensor nodes to reduce the power requirement of communication and increase battery life.

### Future Work

Many applications require downlink communication, i.e., communication from the reader to the tag. Several research groups have investigated various forms of downlink modulation, e.g., amplitude shift keying of the carrier [5], on-off keying [22] and binary frequency-shift keying [16]. Modulation on the carrier poses two challenges: first, the wireless energy transfer may be affected during modulation. Second, the downlink signal must be decoded very simply, as the tags are highly resource-constrained devices, eliminating high sampling rates and complex filters on the tag. In future work, we will investigate different downlink modulation schemes and demodulation and decoding schemes.

An additional challenge arises when multiple tags are charged and read out with a single reader. We will investigate different typical methods for medium access control and multiple access for suitability for acoustic backscatter communication, e.g., TDMA, FDMA or orthogonal codes. Negotiation of suitable carrier frequencies for multiple tags at different locations may also be a challenge in multi-tag environments. Here, investigation of dynamic beamforming may be beneficial to not only increase transmission efficiency but also power and read out specific sensor nodes individually.

Moreover, our experiments, as well as prior work, have been performed in controlled environments. In many possible applications, additional noise sources exist. For example, traffic causes strong vibrations in bridges, or motors introduce strong vibrations over a large spectrum. More work is required to identify the real-world conditions in such scenarios and explore potential mitigation strategies to deal with additional noise.

## 6. Conclusions

In this work, we have reasoned that backscatter communication can be of significant interest in reducing power consumption during communication between sensor nodes embedded in metals. We reviewed current studies for wireless acoustic power transfer and communication using guided lamb waves in metal structures, showing no demonstration of combined power transfer and backscatter communication exists to date.

We present a prototype, give detailed information about the prototype hardware, and investigate the potential of wireless power transfer over larger distances than the existing studies. We showed that a maximum of more than 10 mW can be transferred over 1 m with an efficiency of 8%. Over 3 m distance, efficiency is reduced drastically to only 2.8%, but still, 4 mW could be transferred. The presented harvester requires the source to provide at least 200 μW for successful harvesting, but we discuss potential optimizations to reduce this requirement. For acoustic backscatter communication, we tackle two challenges for real-world deployments of such devices, i.e., (a) self-interference of the backscatter reader limits the distances over which backscatter communication can be used, and (b) low-power devices use inaccurate and unstable oscillators that significantly impede the decoding of messages. The presented prototype showed that by using digital phase-locked loops, we can successfully track the errors induced by the unstable oscillators and decode messages at a data rate of 2 kbit s${}^{-1}$ over 3 m, which was not feasible without tracking mechanisms in place. This enables low-cost and low-complexity implementations of battery-free embedded sensor nodes in closed metal containers for structural and industrial monitoring, which can be constructed from off-the-shelf discrete components.

## Figures and Tables

**Figure 1 sensors-23-03617-f001:**
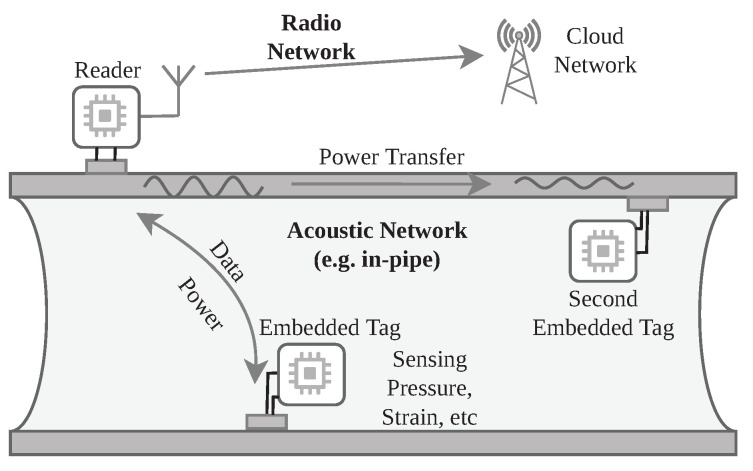
Concept of a potential application, e.g., an acoustic pipeline monitoring network. A reader powers multiple embedded sensor nodes using acoustic-guided waves.

**Figure 2 sensors-23-03617-f002:**
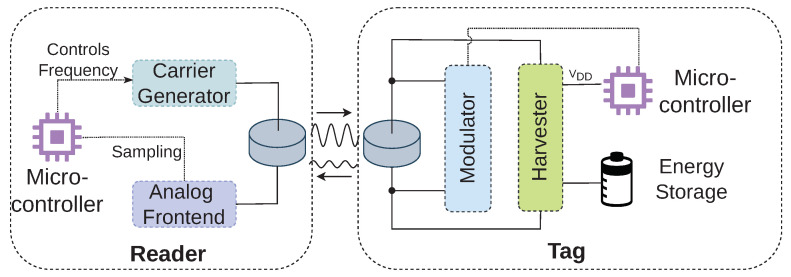
The general architecture of the hardware prototype. The backscatter modulator and the energy harvester work on the same piezoelectric transducer.

**Figure 3 sensors-23-03617-f003:**
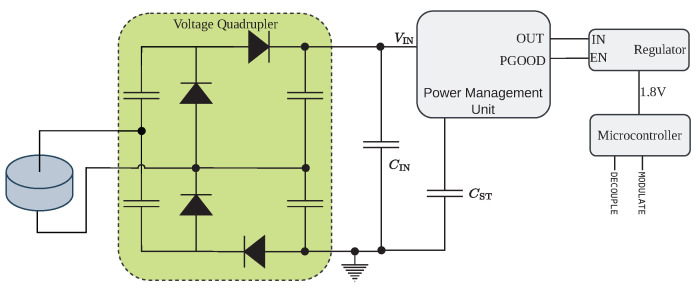
The harvesting circuit connected to the piezoelectric transducer. The voltage quadrupler is adapted from [7].

**Figure 4 sensors-23-03617-f004:**
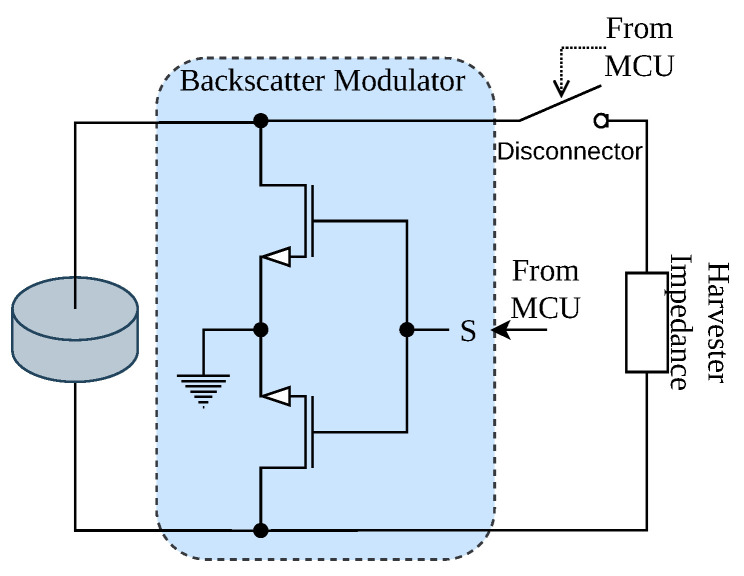
The modulator circuit switches the impedance presented to the piezo between the short-circuit and open-circuit state.

**Figure 5 sensors-23-03617-f005:**
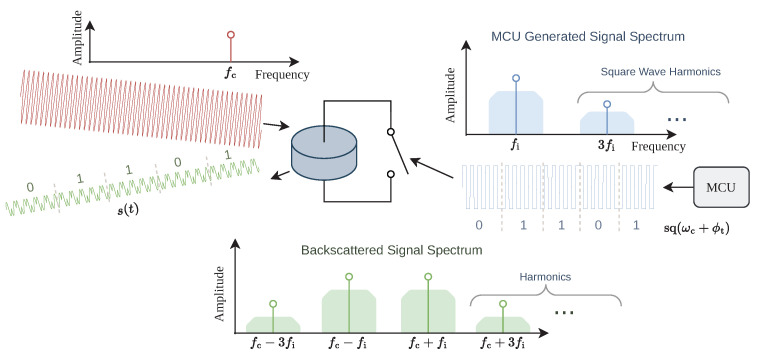
At the tag, the incoming sine carrier wave is modulated with a PSK-modulated square wave at an intermediate frequency from the MCU. While the square wave has infinite bandwidth, the reflected signal also has infinite bandwidth.

**Figure 6 sensors-23-03617-f006:**
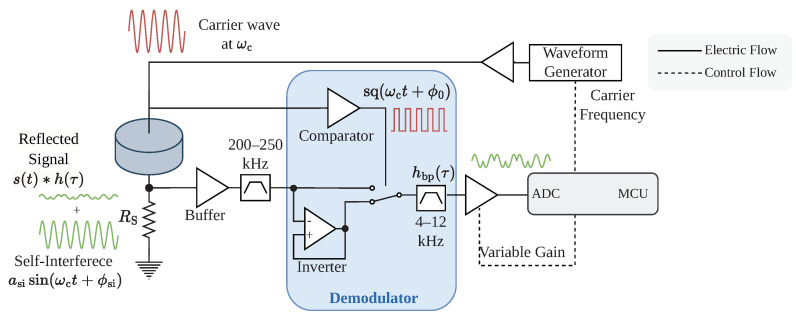
The signal is filtered and demodulated in hardware to remove the carrier frequency at the reader. Afterward, a bandpass filter removes the self-interference, only leaving the backscattered signal.

**Figure 7 sensors-23-03617-f007:**
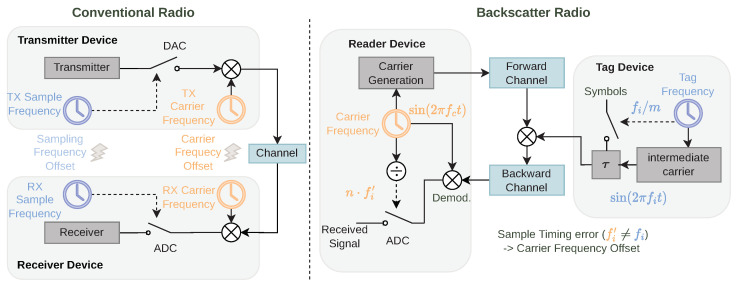
In the backscatter radio, the CFO and Symbol Timing Error (STE) stem from a mismatch in the same pair of clocks.

**Figure 8 sensors-23-03617-f008:**
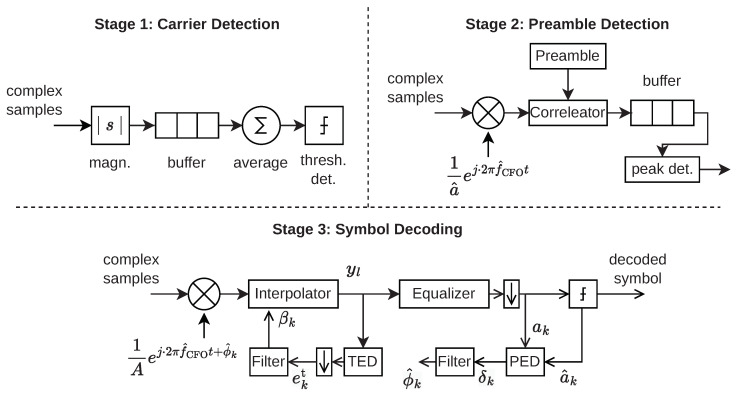
The SDR pipeline has three stages: Carrier detection, preamble detection, and symbol decoding. CFO ${\hat{f}}_{\mathrm{CFO}}$, phase error $\hat{\phi }$, and TE β must be estimated and tracked throughout the message.

**Figure 9 sensors-23-03617-f009:**
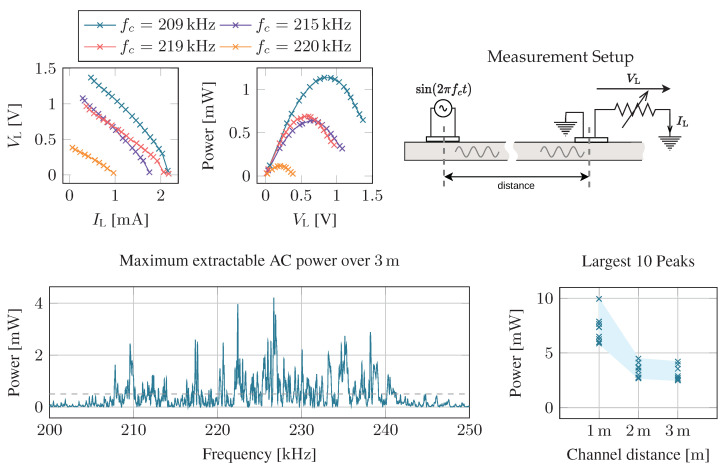
The maximum power point is achieved at half the open circuit voltage. Wirelessly transmitted power is highly frequency dependent, and the distance between transmitter and receiver is important.

**Figure 10 sensors-23-03617-f010:**
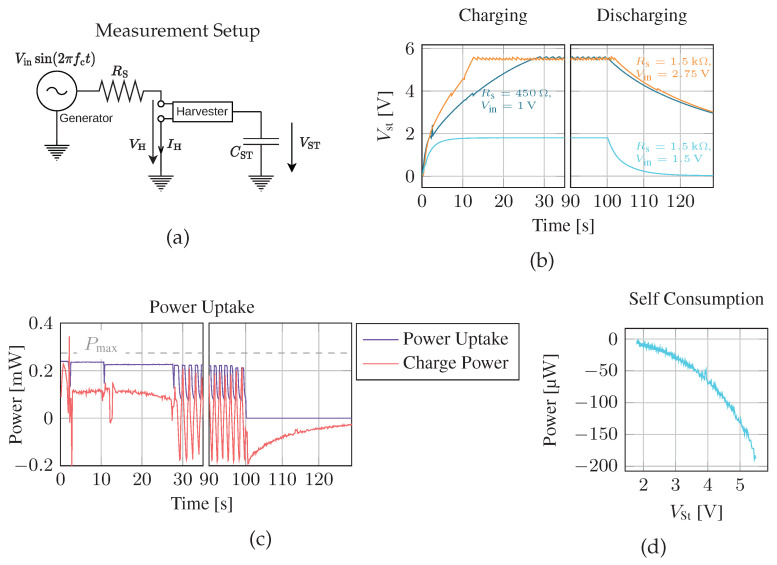
The measurement setup is shown in (**a**), and charge curves (**b**) show the voltage at the storage capacitor over time. The power consumed by the harvester is suboptimal (**c**), and the PMU consumes part of the power itself. The self-consumption is dependent on the voltage at the storage capacitor (**d**).

**Figure 11 sensors-23-03617-f011:**
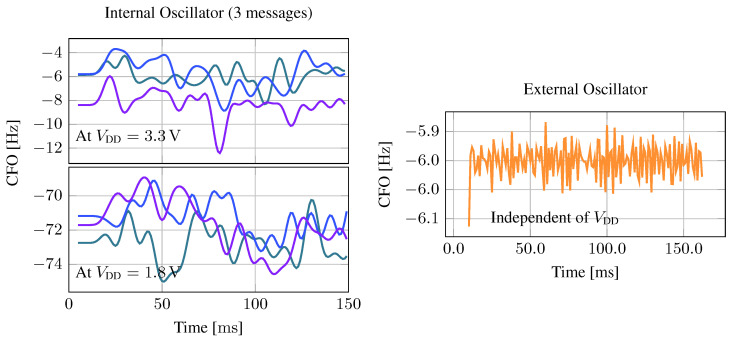
Carrier frequency offset measured throughout three consecutive messages (differently colored lines). The supply voltage level strongly influences the generated carrier frequency when using the internal oscillator, but it also fluctuates significantly during the message. The external oscillator is independent of the supply voltage and much more stable.

**Figure 12 sensors-23-03617-f012:**
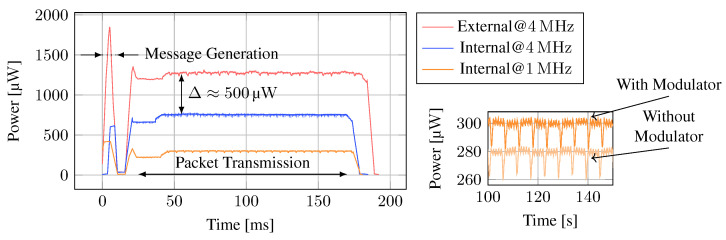
The power consumption of the tag MCU with different oscillators. External crystal oscillators at equivalent frequencies consume significantly more power than the low-power internal oscillator. The backscatter modulation itself only consumes around 20 μW.

**Figure 13 sensors-23-03617-f013:**
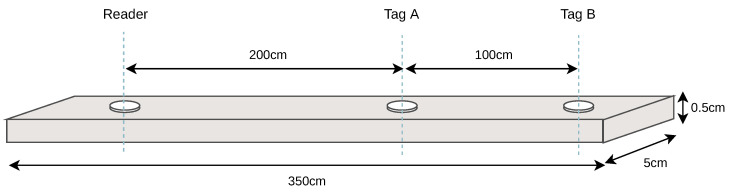
The evaluation setup on a 3.5 m steel beam (not to scale). We use two different channels with lengths: 2 m and 3 m.

**Figure 14 sensors-23-03617-f014:**
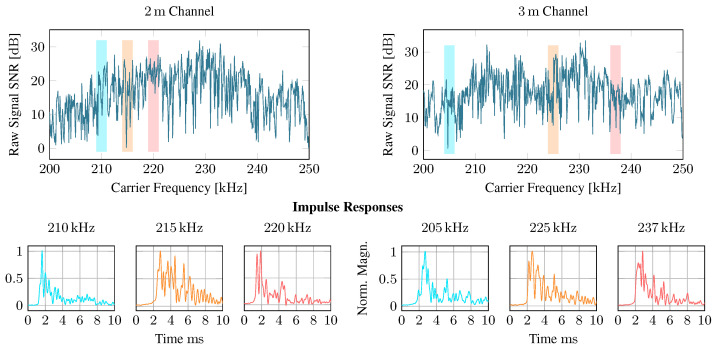
SNR over the raw signal and after equalizing the received messages in 2 and 3 m channels.

**Figure 15 sensors-23-03617-f015:**
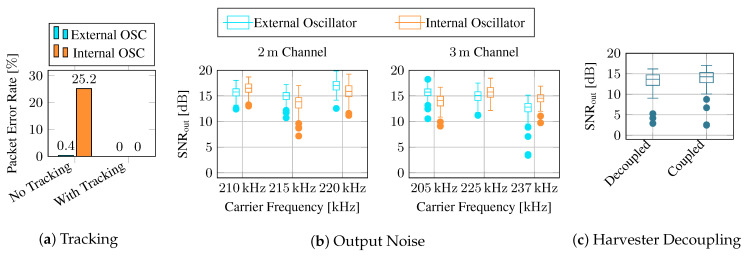
Tracking phase and timing errors with a PLL improves the packet error rate drastically when using the unstable internal oscillator (**a**). When using a PLL, internal and external oscillators perform similarly, and all messages are received and decoded correctly in the channels under test (**b**). The switch used to decouple the harvester from the modulator circuit does not significantly impact output SNR (**c**). All recorded messages are available in Appendix A.

**Figure 16 sensors-23-03617-f016:**
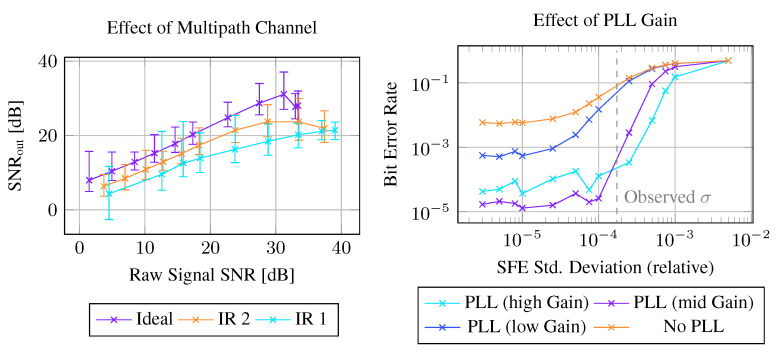
Using simulated signals, the influence of the channel impulse response on the output noise of the decoder is characterized (**left**), where IR1 is the recorded channel impulse response from the 3 m channel at $f_{c}=225$ kHz, and IR2 is from the 2 m channel at $f_{c}=210$ kHz. Furthermore, the PLL gain is an important parameter to choose depending on SFE variability (**right**). The simulation generated signals with an SNR of 15 dB during this evaluation.

**Figure 17 sensors-23-03617-f017:**
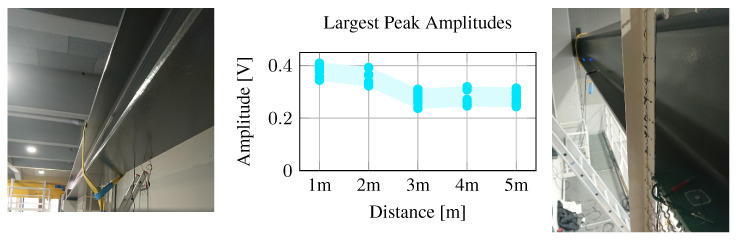
Experiment on a large structure with transducers placed at distances 1, 2, 3, 4, and 5 m. The graph shows the amplitudes of the ten best frequencies over different distances.

**Table 2 sensors-23-03617-t002:** Summary of existing studies on lamb-wave-based active communication studies compared to the presented backscatter communication study.

Study	Spec.	Distance m	Frequency kHz	Modu Lation	Data Rate kbit s${}^{-1}$	ISI Mitigation ^1^	Impl. Level
Tang, 2022 [5]	Plate (CFRP)	<0.3	400	OOK	15	None	CMOS IC
Bahouth, 2022 [22]	Plate (Alum.)	0.08	200	BPSK	350	CCIR	Lab Equipment
Sun, 2021 [18]	Plate (Alum.)	0.4	500	OOK	100	CCIR & MD	Lab Equipment
Heifetz, 2021 [23]	Pipe	1.8	2000	ASK	2	None	Lab Equipment
Jin, 2013 [24]	Pipe	1.5	250	PPM	10	TR	Lab Equipment
**This work**	Beam (Steel)	3	225	PSK	2	Equalizer	MCU + AF ^2^

^1^ CCIR: cross-correlation with impulse response, MD: mechanical damping, TR: time reversal; ^2^ AF: Custom Analog Frontend.

## Data Availability

Research data is partly uploaded as Appendix A.

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
