# Peer review of "Acoustic Backscatter Communication and Power Transfer for Batteryless Wireless Sensors"

_sensors, 2023, doi:10.3390/s23073617_

Round 1
Reviewer 1 Report
In this paper the authors have reviewed the available approaches to acoustic WPT and active and passive acoustic through-metal communication. And also proposed a batteryless and backscattering tag prototype from commercially available components.
The paper is well structured. But there are some observations as follows.
1. ‘Acoustic wireless power transfer (WPT) and piezoelectric backscatter communication
along metal surfaces enable batteryless sensors in shielded and hard-to-reach places for structural and industrial health monitoring’…. Hard to understand.
2. “Simulation and real-world measurements show that using DPLLs enables reliable backscatter communication with inaccurate clocks, achieving communication at 2 kB s-1 over 3 m.” ….rewrite the sentence
3. Page 2 line 76: ‘we employ algorithms”…..Be more specific as this presents your contribution.
4. The authors have presented the system modeling through several block diagrams and related discussion. But it will be more attractive if it includes detailed mathematical models/expression with suitable expression.
5. Fig 8. Several processes are there but there is no mathematical modeling. A complete and systematic model will help the readers to understand the paper in a better way.
6. Include future scope of work in the conclusion section.
7. Page 14 line 463: “However, further optimization of the PMU might allow less leakage and enable more effective harvesting from low-power sources.”……Elaborate it. What do mean by further optimization?
8. Equations are presented without any proper definition of the variables that are used e.g (7)
9. “In the performed experiments, the efficiency of the harvester is around 40 % to 50 %,”…Include all the related data.
Author Response
We thank the reviewer for the valuable suggestions and positive feedback on our manuscript.
We have added several new passages in the manuscript and provided further suggested details in the appendix. For easy revision, we colored new and modified passages in the revised manuscript (see attachment) and added a label at each modification referring to the comment we addressed. Additions are colored in green, and modifications are colored in blue. As an example, the label (Rev1.3) refers to the third comment given by reviewer 1.
We believe the revised manuscript is greatly improved and hope that the reviewer will find that all their comments are appropriately addressed.
In the remainder, we give point-by-point responses to all comments.
- 'Acoustic wireless power transfer (WPT) and piezoelectric backscatter communication along metal surfaces enable batteryless sensors in shielded and hard-to-reach places for structural and industrial health monitoring'…. Hard to understand.
Thank you for that suggestion, we rewrote the sentence for more clarity.
- "Simulation and real-world measurements show that using DPLLs enables reliable backscatter communication with inaccurate clocks, achieving communication at 2 kB s-1 over 3 m."….rewrite the sentence
Thank you, we split this into two sentences.
- Page 2 line 76: 'we employ algorithms"…..Be more specific as this presents your contribution.
The reviewer correctly pointed out that we did not yet specify the specific algorithms in the introduction. We clarified that we use digital phase-locked loops to track phase offset and sample timing mismatch with a derivative timing error detector.
Comments 4 and 5 are addressed together:
- The authors have presented the system modeling through several block diagrams and related discussion. But it will be more attractive if it includes detailed mathematical models/expression with suitable expression.
- Fig 8. Several processes are there but there is no mathematical modeling. A complete and systematic model will help the readers to understand the paper in a better way.
We agree with the reviewer, that a mathematical model of the signal-processing steps improves the manuscript. However, parts of that model are very cumbersome, and we believe it impairs readability. Therefore, we deferred some additions to the appendix. Specifically, we modeled the backscattered signal on page 4 and added the new notation to Figure 5.
Proof showing the effectiveness of the presented modulator circuit in Figure 6 has been added to Appendix A. We also added additional terms to Figure 6, where appropriate.
Finally, we described the equations used by the PLLs to track and detect Phase and Timing errors on page 12. Appendix B further explains the derivative timing error detector for readers unfamiliar with the topic.
- Include future scope of work in the conclusion section.
Future work has been detailed in section 5.1.
- Page 14 line 463: "However, further optimization of the PMU might allow less leakage and enable more effective harvesting from low-power sources."……Elaborate it. What do mean by further optimization?
Our comment was related to the significant reduction of leakage current by choosing larger storage capacitors with lower voltages, as mentioned in the previous paragraph. We acknowledge that this was unclear in our manuscript and changed the manuscript for clarity (page 15 in the revised manuscript).
- Equations are presented without any proper definition of the variables that are used e.g (7)
The reviewer is correct. We added variable definitions for the mentioned equation and re-checked the remaining equations in the manuscript.
- "In the performed experiments, the efficiency of the harvester is around 40 % to 50 %,"…Include all the related data.
More specific statements about efficiencies can not be made. Efficiency is highly dependent on the power a source can deliver, its voltage, and even the storage capacitor's voltage. For example, Figure 10 c) shows the maximum available power for one source and the harvested power over time. The efficiency drops over time as the storage capacitor charges. In cases where the source provides only tens of microwatts, the harvester's efficiency drops close to zero because it consumes most of the available power itself. We mentioned "40 to 50 percent" as a ballpark figure to characterize the harvester when sufficient power is available at the source. We also explained this in the revised manuscript (page 15) to clarify this for all readers.
Reviewer 2 Report
A. General idea of the presented scientific work
The article concerns the practical implementation of wireless energy transfer using mechanical vibrations.
The publication concerns a sophisticated method of energy transfer that can be used in specific conditions. The use of mechanical vibrations is currently a rarely used method due to the great difficulties in implementation.
My overall assessment of the article is positive due to the comprehensive, exhaustive description of the issue, discussion on the construction of the transmitter and receiver, and experimental verification.
B. Assessment of methodology
I assess correctly the methodology of the research.
The discussion of the other works is comprehensive. It is a valuable part of the paper.
The basic elements of the theory have been presented. The mathematical formalism is simple. The authors do not present theory of the method. The analysis of mechanical vibrations boils down to the basic relationships that have been described.
The strength of the paper is the practical part: elaboration of the elements of the test stand, actuators, measurements. The validation of the proposed method was carried out on the basis of the experiment data.
The completeness of the description and the practical effect of the research are additional factors in favor of a good review of the paper.
C. Form, layout of the paper
The general structure and layout of the paper is correct (including body of the text, figures, tables, and equations). The implemented notation is correct and consistent. Only the notation in eq. (1) should be modified. The scalar variables (real or complex) should be written in italics.
The list of publications has 24 items (incl. 2 references to authors' papers).
The language of the paper is correct.
D. Some questions / critical comments on the work.
1. Energy transfer efficiency is very low, not exceeding 3%. Would the application (sputtering) of additional elements (a kind of waveguides) not help in directional energy transfer?
2. Energy transfer efficiency can be improved if:
a) the system operates in mechanical resonance;
b) there is a wave matching of the transmitter and the receiver.
Is it possible to include tuning elements in the proposed method (in order to obtain resonance and wave matching).
3. The condition of transfer of mechanical wave can be change (e.g. additional mass connected to the system). Does the proposed method can adjust the bias point of the system to current condition of working.
E. Final opinion
In my opinion the scientific level of the presented solution is good enough (acceptable rank).
The answer to my comments (three points with some remarks) can improve the quality of the paper and deliver more information for readers.
Author Response
We thank the reviewer for the valuable suggestions and positive feedback on our manuscript.
We have added several new passages in the manuscript and provided further suggested details in the appendix according to points raised by all three reviewers. For easy revision, we colored new and modified passages in the revised manuscript (see attachment) and added a label at each modification referring to the particular comment we addressed. Additions are colored in green, and modifications are colored in blue. As an example, the label (Rev1.3) refers to the third comment given by reviewer 1.
We believe the revised manuscript is greatly improved and hope the reviewer finds that we addressed all their comments appropriately.
In the remainder, we give point-by-point responses to all comments.
- The notation in eq. (1) should be modified. The scalar variables (real or complex) should be written in italics.
Thanks for spotting that mistake. We adapted the notation.
In the following, we want to address points 1 to 3 jointly.
- Energy transfer efficiency is very low, not exceeding 3%. Would the application (sputtering) of additional elements (a kind of waveguides) not help in directional energy transfer?
- Energy transfer efficiency can be improved if:
- a) the system operates in mechanical resonance;
- b) there is a wave matching of the transmitter and the receiver.
Is it possible to include tuning elements in the proposed method (in order to obtain resonance and wave matching).
- The condition of transfer of mechanical wave can be change (e.g. additional mass connected to the system). Does the proposed method can adjust the bias point of the system to current condition of working.
The reviewer is correct. The overall low efficiency can be improved in several waves, as suggested. We did not apply the suggested measures because they compromise with our goal of generality, simple and versatile deployment, and cheap, general-purpose devices; however, we think this is worthwhile to discuss these points. Therefore, as summarized in the following paragraphs, we added paragraphs to the discussion and future work section.
First, directional waves can concentrate transmitted energy in the general direction of the receiver. In some prior work, rectangular transducers have been used to generate planar waves traveling in one direction [20]. We neglected this approach as we envision multi-tag environments in the future, where one reader can power multiple tags.
Others investigated transducer arrays to implement beamforming [17]. While this allows dynamically focusing the energy on specific tags, it increases the complexity and deployment cost of the reader.
Second, the efficiency increases significantly when the whole system works in resonance. However, the resonance of the structure may be a highly complicated function depending on geometry and boundary conditions. Tseng et al. [15] used a finite element model simulation of the metal waveguide to find its resonance frequencies, chose transducers with matching resonance, and determined their optimal placement position to achieve the highest transfer efficiency. We wanted to design a flexible solution usable on various structures without individual adaptations. We envision placing tags and readers in a plug-and-play-like manner. Moreover, resonance is affected by changing conditions (e.g., temperature), so maintaining resonance in practice is hard, if not infeasible.
For the same reason, we did not consider actively adapting the structure, e.g., with added mass, to shift the resonance to a desired frequency.
Once again, we thank the reviewer for their constructive feedback and assessment. We are confident that we have addressed the raised concerns and improvement suggestions.
Reviewer 3 Report
The use of small and inexpensive sensors in various domains has increased with the growth of IoT. However, sensors located in hard-to-reach places cannot be easily accessed for battery replacement or wired connections. To address this issue, one solution is to use RF power, but in RF shielded environments, transmitted RF power cannot reach the sensor node where power can be harvested for sensor use. Acoustic wireless power transfer (WPT) and piezoelectric backscatter communication are two promising technologies that enable batteryless sensors in hard-to-reach places with RF shielding. In this paper, the authors investigate available approaches to acoustic WPT and active/passive acoustic communication through metal, and their research is unique in exploring passive ground wave communication (GWC) systems that use backscatter for sensor communication, whereas most previous research has focused on active systems that use more power and components.
The authors have successfully implemented a batteryless and backscattering tag prototype that uses off-the-shelf components to bridge the gap between practical use and laboratory experiments. However, one of the main challenges in using commercially available parts is their tendency to yield unstable and inaccurate oscillations. The authors have overcome this challenge by using a software-defined radio (SDR) and digital phase-locked loop (DPLL) to ensure reliable backscatter communication with inaccurate clocks.
Overall, the draft paper is well-written and informative, with clear and orderly presentation of the problem, background, methodology, and interpretation of results. However, the tests were conducted in controlled environments, and the authors did not explicitly address the potential impact of environmental noise, such as vibrations in a work environment, on the received data. Future work could benefit from discussing potential ways to mitigate the impact of environmental noise on communication performance.
Author Response
We thank the reviewer for the valuable suggestions and positive feedback on our manuscript.
We have added several new passages in the manuscript and provided further suggested details in the appendix according to points raised by all three reviewers. For easy revision, we colored new and modified passages in the revised manuscript (see attachment) and added a label at each modification referring to the particular comment we addressed. Additions are colored in green, and modifications are colored in blue. As an example, the label (Rev1.3) refers to the third comment given by reviewer 1.
We believe the revised manuscript is greatly improved and hope the reviewer finds that we addressed all their comments appropriately.
In the remainder, we give point-by-point responses to all comments.
- Overall, the draft paper is well-written and informative, with clear and orderly presentation of the problem, background, methodology, and interpretation of results. However, the tests were conducted in controlled environments, and the authors did not explicitly address the potential impact of environmental noise, such as vibrations in a work environment, on the received data. Future work could benefit from discussing potential ways to mitigate the impact of environmental noise on communication performance.
The reviewer is correct. Some applications, e.g., structural health monitoring on bridges or industrial monitoring on heavy machines, suffer from severe noise. Real-world tests are required to assess the limitations in such scenarios. We took the reviewer's objection up in the future work section.
Round 2
Reviewer 1 Report
The authors have significantly improved the paper.